# Learning Temporal Abstraction with Information-theoretic Constraints for Hierarchical Reinforcement Learning

## Abstract

Applying reinforcement learning (RL) to real-world problems will require reasoning about action-reward correlation over long time horizons. Hierarchical reinforcement learning (HRL) methods handle this by dividing the task into hierarchies, often with hand-tuned network structure or pre-defined subgoals. We propose a novel HRL framework TAIC, which learns the temporal abstraction from past experience or expert demonstrations without task-specific knowledge. We formulate the temporal abstraction problem as learning latent representations of action sequences and present a novel approach of regularizing the latent space by adding information-theoretic constraints. Specifically, we maximize the mutual information between the latent variables and the state changes. A visualization of the latent space demonstrates that our algorithm learns an effective abstraction of the long action sequences. The learned abstraction allows us to learn new tasks on higher level more efficiently. We convey a significant speedup in convergence over benchmark learning problems. These results demonstrate that learning temporal abstractions is an effective technique in increasing the convergence rate and sample efficiency of RL algorithms.

## 1 Introduction

Reinforcement learning (RL) has been successfully applied to many different tasks (Mnih et al., 2015; Zhu et al., 2017). However, applying it to real-world tasks remains a challenging problem, mainly due to the large search space and sparse reward signals. In order to solve this, many research efforts have been focused on the hierarchical reinforcement learning (HRL), which decomposes an RL problem into sub-goals. By solving the sub-goals, low-level actions are composed into high-level temporal abstractions. In this way, the size of the searching space is decreased exponentially. However, the HRL often requires explicitly specifying task structures or sub-goals (Barto & Mahadevan, 2003; Arulkumaran et al., 2017). How to learn those task structures or temporal abstractions automatically is still an active studying area.

Many different strategies are proposed for automatically discovering the task hierarchy or learning the temporal abstraction. Some early studies try to find sub-goals or critical states based on statistic methods (Hengst, 2002; Jonsson, 2006; Kheradmandian & Rahmati, 2009). More recent work seeks to learn the temporal abstraction with deep learning (Florensa et al., 2017; Tessler et al., 2017; Haarnoja et al., 2018a). However, many of these methods still require a predefined hierarchical policy structure (e.g. the number of sub-policies), or need some degree of task-specific knowledge (e.g. hand-crafted reward function).

We present a general HRL framework TAIC (Temporal Abstraction with Information-theoretic Constraints), which allows an agent to learn the temporal abstraction from past experiences or expert demonstrations without task-specific knowledge. Built upon the ideas of options framework (Sutton et al., 1999) and motor skills (Lin, 1993), we formulate the temporal abstraction problem as learning a latent representation of action sequences. In order to obtain good latent representations, we propose a novel approach to regularize the latent space by using information-theoretic constraints. The learned abstract representations of action sequences (we called options) allow us to do RL at a higher level, and easily transfer the knowledge between different tasks.

Our contributions are: 1) We formulate the temporal abstraction problem as learning a latent representation of action sequences. Motivated by works using Recurrent Variational AutoEncoders (RVAE) to model sequential data in neural language processing (NLP) and other areas (Bowman et al., 2015; Ha & Eck, 2017), we employ RVAE to perform temporal abstraction in RL. 2) We propose a regularization approach on the option space. It constrains the option to encode more information about its consequence (how the option changes the states). We present both theoretical derivations and practical solutions. 3) We show in the experiments that our learned temporal abstraction conveys meaningful information and benefit the RL training. In addition, the proposed framework provides an efficient tool for transferring knowledge between tasks.

## 2 RELATED WORK

HRL is a long-standing study topic. Schmidhuber (1990) proposed HRL with temporal abstraction in early 1990s. Schmidhuber (1991) proposed a system in which recurrent NNs generate sequences of sub-goals, a evaluator NN predicts the rewards of going from start to goal and a RL machine tries to use such sub-goal sequences to achieve final goals (Schmidhuber & Wahnsiedler, 1993). The options framework is a popular formulation for considering the problem with a two-level hierarchy (Sutton et al., 1999). A sequence of primitive actions are converted into options. The traditional Markov Decision Process (MDP) problem is extended into semi-MDP with the use of options. Parr developed an approach called Hierarchies of Abstract Machines to calculate hierarchically structured MDP policies (Parr & Russell, 1998). Dietterich proposed MAXQ method, which performs value function decomposition over a given task structure (Dietterich, 2000). These early work assume the task hierarchy is predefined by human experts.

For the automatic task decomposition problem, many methods try to find sub-goals or critical states based on statistic methods (Hengst, 2002; Jonsson, 2006; Kheradmandian & Rahmati, 2009). These methods cannot handle continuous control problem. More recent work seeks to learn the temporal abstraction with deep learning (Florensa et al., 2017; Tessler et al., 2017; Vezhnevets et al., 2017; Haarnoja et al., 2018a; Nachum et al., 2018). Frans et al. (2017) developed a two-layer hierarchy of policies, including one meta-policy and several primitive policies. Andreas et al. (2017) defined the policy sketches, which annotate tasks with sequences of sub-tasks, and learn the sub-tasks and upper-level tasks jointly. Kulkarni et al. (2016) presented hierarchical-DQN, which integrates hierarchical action-value functions with intrinsic reward based on predefined sub-goals. Bacon et al. (2017) combined the options framework with policy gradient and proposed an option-critic architecture. These methods either require a predefined hierarchical policy structure (e.g. the number of sub-policies) or need to specify sub-goals. We propose a framework that allows learning temporal abstraction without predefined sub-goals.

The idea of learning latent representation for HRL has been proposed before (Eysenbach et al., 2018; Hausman et al., 2018), but they did not learn from action sequences. The SeCTAR algorithm learns a latent representation from from state sequences in trajectories using RVAE (Co-Reyes et al., 2018). Macro-action methods share the similar idea of combining sequences of actions into options (Hauskrecht et al., 1998; Vezhnevets et al., 2016). However, existing literature only *combines* primitive actions, while our work learns an *abstraction*, which we will show later is more beneficial for HRL training. Our work is related to Fabius & van Amersfoort (2014); Bowman et al. (2015); Ha & Eck (2017), since they also utilize RVAE to encode sequential data. In this paper, we apply RVAE to model the abstraction of action sequences.

## 3 APPROACH

### 3.1 PROBLEM FORMULATION

We consider a MDP problem $M = (\mathcal{S}, \mathcal{A}, P, R, \gamma)$, where $\mathcal{S}$ is the set of states, $\mathcal{A}$ is the action set, $P : (\mathcal{S}, \mathcal{A}) \mapsto \mathcal{S}$ is the transition model, $R : (\mathcal{S}, \mathcal{A}) \mapsto \mathbb{R}$ is the reward function, $\gamma$ is the discount factor. Sutton et al. (1999) presents the option framework, where an option is formulated as a tuple $(\mathcal{I}, \pi, \beta)$ in which $\mathcal{I} \in \mathcal{S}$ is the initiation set, $\pi$ is the sub-policy and $\beta : \mathcal{S} \mapsto [0, 1]$ is the termination condition. The option framework and its followers (Bacon et al., 2017; Kulkarni et al., 2016) either manually designed or learned a *finite* set of options. The number of option becomes a

hyperparameter, and each option in the set is usually *independent* with one another. We propose the TAIC framework, in which we model the option $o$ as a continuous random variable, which denotes the latent representation of a sequence of actions. Furthermore, following the option framework we define $(\mathcal{I}(o), \pi(o), \beta(s, o))$ as the function of the random variable $o$. $\mathcal{I}(o)$ is the initiation condition, which is assumed to be the entire state space $\mathcal{S}$ for simplification. We define the function $\pi(o)$ as the sub-policy that maps the latent variable $o$ to a sequence of actions. The termination condition $\beta(s, o)$ controls the temporal length of the option, and will be detailed later.

This modification of the option definition brings important difference. In the original option framework, each option represents one sub-policy. Above those sub-policies, there is a meta-policy $\pi_\Omega$ that acts as a classifier, choosing one of the sub-policies at a longer time horizon. In contrast, the TAIC framework specifies the option $o$ as a continuous random variable, which could have an infinite number of values. Sub-policy is defined as a function over the random variable. Our meta-policy $\pi_\Omega$ outputs directly the random variable $o$. This changes the framework from discrete options to continuous options, and allows us to put constraints on the option space. So that the options with similar consequences become closer in the option space.

Given a set of past experiences $\Lambda = \{\tau_0, \tau_1, ..., \tau_m\}$, where $\tau_i \in \Lambda$ is one trajectory $\{s_0, a_0, r_0, s_1, a_1, r_1, ..., s_k, a_k, r_k\}$, which comes from the agent's past experience or expert demonstration, our problem is to learn a temporal abstraction $o$, and the corresponding $(\mathcal{I}(o), \pi(o), \beta(s, o))$, so that the $o$ should be able to apply to the RL task, and improve the training efficiency.

## 3.2 TEMPORAL ABSTRACTION AS LEARNING LATENT REPRESENTATION

We consider the problem of learning latent representations $o$ from action sequences $\{a_{0...k_0}, a_{0...k_1}, ...\}$. To model the sequential data with variable length, we use the recurrent auto-encoder (AE), which has been used intensively in natural language processing (NLP) (Bowman et al., 2015) and other sequential modeling tasks (Ha & Eck, 2017). Specifically, we deploy recurrent variational auto-encoders (RVAE) (Kingma & Welling, 2013; Fabius & van Amersfoort, 2014), because it is empirically shown to be better at finding hidden features of the inputs (Chung et al., 2015).

In general, we would like to calculate the posterior $p(o|a_{0...k})$, where the option $o$ captures the intrinsic features of the action sequences. The RVAE solves this by approximating the true posterior by $q(o|a_{0...k})$ and then optimizing a lower bound on the log-likelihood (Fabius & van Amersfoort, 2014). The log-likelihood of an action sequence $a_{0...k}$ can be written as:

$$\log p(a_{0...k}) = KL(q(o|a_{0...k})||p(o|a_{0...k})) + \mathcal{L}(o) \tag{1}$$

where $KL(p|q)$ is the Kullback-Leibler divergence, and the $\mathcal{L}(o)$ is the evidence lower bound:

$$\mathcal{L}(o) = -KL(q(o|a_{0...k})||p(o)) + \mathbb{E}_{q(o|a_{0...k})}[\log p(a_{0...k}|o)] \tag{2}$$

In order to find a good approximation of $p(o|a_{0...k})$ with $q(o|a_{0...k})$, namely minimize $KL(q(o|a_{0...k})||p(o|a_{0...k}))$, we need to maximize the evidence lower bound $\mathcal{L}(o)$. The conditional distributions $q(o|a_{0...k})$ and $p(a_{0...k}|o)$ are approximated by two networks, a recurrent encoder $E : a_{0...k} \mapsto o$ and a recurrent decoder $D : o \mapsto a_{0...k}$. Following the common VAE formulation, we model $q(o)$ with the Gaussian distribution, and the training objective is to minimize a reconstruction loss and a KL loss (Kingma & Welling, 2013):

$$L_{rvae} = L_{KL} + L_{Recons} \tag{3}$$

Specifically, the encoder $E$ takes in an action sequence $a_{0...k}$, and outputs the option represented by two vectors $o_\mu$ and $o_\sigma$, namely the mean and standard deviation in the Gaussian distribution. The $L_{KL}$ is the KL divergence between this output and the prior Gaussian distribution $\mathcal{N}(0, 1)$. The decoder $D$ takes in the sample of option $o$, and outputs $\hat{a}_{0...k}$. The $L_{Recons}$ is defined as the mean-square-error between $a_{0...k}$ and $\hat{a}_{0...k}$.

## 3.3 INFORMATION-THEORETIC CONSTRAINTS IN THE LATENT SPACE

The reconstruction loss in the RVAE setting suggests it encodes the action sequences with respect to the L2 distance in action space. Figure 1 shows a simple example in 2D navigation task. Two

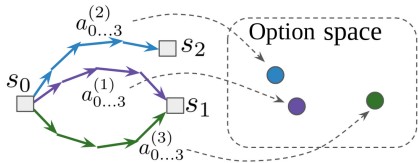

Figure 1: Using RVAE, action sequences are encoded w.r.t the L2 distance in action space. As a result, although $a_{0...3}^{(1)}$ and $a_{0...3}^{(3)}$ have exactly the same consequence, their codes will be far away in the option space. This is not a desired property.

action sequences $a_{0...3}^{(1)}$ and $a_{0...3}^{(2)}$ are with the same length, and only have small difference in each step of action. Due to the error compounding, the two sequences end up in very different resulting states $s_1$ and $s_2$. However, $a_{0...3}^{(1)}$ and $a_{0...3}^{(3)}$ have big L2 distance in action space, but they have the same consequence (both move from state $s_0$ to $s_1$). In favor of upper-level task solver, they should be encoded closely together in the option space. Imaging the navigation task in real life, in order to reach the door you could follow different paths (avoiding some dynamic obstacles). Those different sequences all have the same coding in your brain "go reach the door".

In contrast to precisely reconstructing the action sequences, our goal is to extract the latent variable capturing the information which could benefit RL training. Intuitively speaking, the option should encode the consequence of action sequence, namely how the sequence changes the state. Formally, we maximize the mutual information between the option and state changes. We decompose it into two terms. The first term is to maximize the mutual information between the option and two states (states before and after a sequence of actions):

$$\max_{o} I(o; s, s') \tag{4}$$

On the other hand, the option should encode the state change, and be decoupled from particular start and end states. For example, navigating to the door from this room should be represented similarly as navigating to the door at another room. This is formulated as minimizing the mutual information between the option and the individual start and end state:

$$\min_{o}[I(o; s) + I(o; s')] \tag{5}$$

Following the definition of the mutual information and the chain rule of the conditional entropy, Equation 4 is transformed into minimizing the summation of two conditional entropy:

$$\begin{aligned} \max_{o} I(o; s, s') &= \max_{o}\{H(o) - H(o|s, s')\} \\ &= \max_{o}\{H(o) - H(o, s, s') + H(s, s')\} \\ &= \max_{o}\{H(o) - (H(s'|s, o) + H(s|o) + H(o))\} \\ &= \max_{o}\{-H(s'|s, o) - H(s|o)\} \\ &= \min_{o}\{H(s'|s, o) + H(s|o)\} \end{aligned} \tag{6}$$

Similarly, Equation 5 is equivalent to maximizing the conditional entropy of $s/s'$ given $o$:

$$\begin{aligned} \min_{o}\{I(o; s) + I(o; s')\} &= \min_{o}\{(H(s) - H(s|o)) - (H(s') - H(s'|o))\} \\ &= \max_{o}\{H(s|o) + H(s'|o)\} \end{aligned} \tag{7}$$

Notice Equation 6 and Equation 7 conflicts with each other, which is also very intuitive. Forcing the option to be irrelevant to the start and end state would make it harder to encode the information of state changes. This is a trade-off depending on how much you want the code to be state-independent. Combining Equation 6 and Equation 7, we obtain our final constraints:

$$\min_{o} H(s'|s, o) = \max_{o} \mathbb{E}[\log p(s'|s, o)] \tag{8}$$

$$\max_{o}\{H(s|o) + H(s'|o)\} = \min_{o}\{\mathbb{E}[\log p(s|o)] + \mathbb{E}[\log p(s'|o)]\} \tag{9}$$

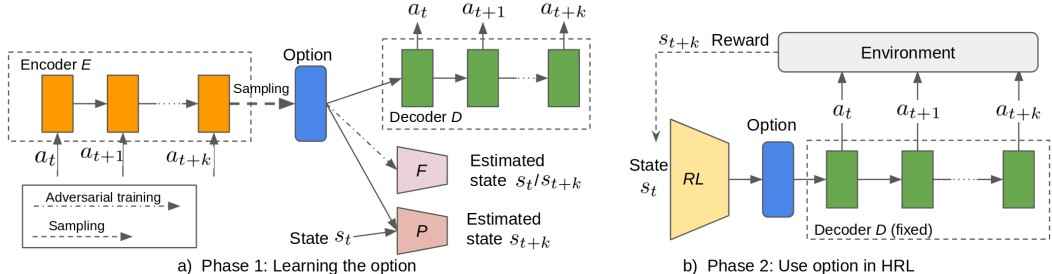

Figure 2: System architecture. $E$, $D$, $F$, $P$ and $RL$ are implemented with neural networks. The gradients of three losses $L_{rvae}$, $L_{adv}$ and $L_{pred}$ will be back-propagated through the encoder $E$, and regularize the option. Note the gradient of $L_{adv}$ will change sign before back-propagating to $E$.

Those two constraints are also approximated with neural networks. For Equation 8, we utilize a predictor network $P : (s, o) \mapsto s'$, which predicts the end state $s'$ given the start state $s$ and the option $o$. To optimize the constraint, we minimize the predictive loss $L_{pred}$, and backpropagate the gradients into the encoder $E$. In order to *minimize* the log-likelihood in constraint Equation 9, we borrow the technology from the adversarial training (Ganin et al., 2016; Shrivastava et al., 2017). We utilize another network $F : o \mapsto (s, s')$, which tries to recover the start state and the end state given the option code. The encoder $E$ and the state estimator $F$ are trained competitively with each other. $F$ acts like a discriminator, trained in minimization of the loss of estimating the states $L_{adv}$, while $E$ is trained with the opposite gradients that try to maximize the loss $L_{adv}$. Thus, $E$ will be pushed to encode option in a way that $F$ is not able to recover the start and end states. In this way, the encode $E$ is regularized by these extra gradients. In practise, all of these networks are trained jointly on the experience set $\Lambda = \{\tau_0, \tau_1, ..., \tau_m\}$. As Figure 2 shows, we have 4 networks ($E$, $D$, $F$ and $P$) collaborating and competing (note $RL$ is not updated in the option learning process). The learned latent representation $o$ is a temporal abstraction that captures the execution consequence of the action sequences.

## 3.4 TERMINATION CONDITION

After learning the option, we can train the agent at a higher level. The HRL policy outputs an option, which is then decoded to a sequence of actions by the decoder $D$. In order to apply the learned option to HRL training, we need to consider the termination condition $\beta(s, o)$. We implement and compare three different methods: Fix-length, Term-output and Term-predict.

### 3.4.1 OPTION WITH FIXED LENGTH

In the simplest setting, we learn the option from the action sequence with fixed length $N$. The decoder $D$ terminates the output after $N$ actions, and then the HRL policy will output another option. This termination condition (referred as Fix-len later) does not depend on the option and state.

This straightforward method is easy to implement. We will show in the experiment section that for most tasks, even this naive implementation would bring benefit to the HRL training.

### 3.4.2 OPTION WITH TERMINATION OUTPUT

In order to learn the option code of arbitrary sequence length, we add another output to the decoder $D$. At each time step $t$, $D$ outputs an action $a_t$ and a termination signal $c_t$, which determines whether this is the last action in the sequence (Figure 3 (a)). The termination output is a 2-class classifier, trained with supervised cross-entropy loss. Note that this termination condition (referred to as Term-output) is only dependent on the option.

Allowing the RVAE to encode various lengths of sequence provides the HRL with more choices. For example, in states that are stable and safe, the HRL can choose longer sequences; while in unstable

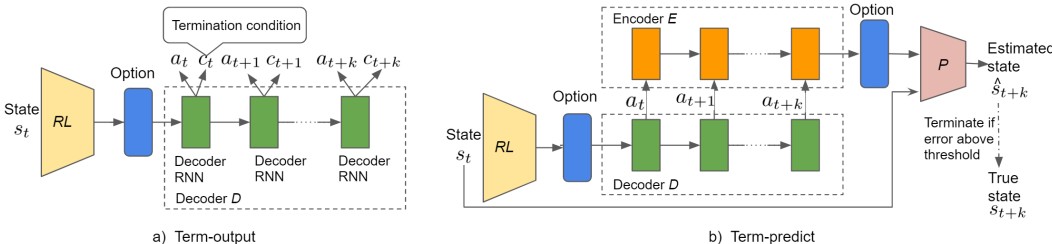

Figure 3: Termination conditions. a) The decoder $D$ outputs a termination signal $c_t$ at each time step $t$. b) The option terminates when the predictor $P$ fails to predict the next state.

---

**Algorithm 1** Train option with unexpectedness condition

---

**Input**:Trajectory $\tau_i = \{s_0, a_0, s_1, a_1, ..., s_k, a_k\}$, networks $E$, $D$, $P$
**Parameter**: Prediction threshold $\delta$
**Output**: Option $o$

1: Let $i = 0$, $L_{pred} = 0$.
2: **while** $L_{pred} < \delta$ **do**
3:     $o_i := E(a_{0...i})$
4:     $\hat{s}_{i+1} := P(s_0, o_i)$
5:     $L_{pred} := MSE(s_{i+1}, \hat{s}_{i+1})$
6:     backpropagate $L_{pred}$
7:     $i := i + 1$
8: **end while**
9: $\hat{a}_{0...(i-1)} := D(o_{i-1})$
10: $L_{Recons} = MSE(\hat{a}_{0...(i-1)}, a_{0...(i-1)})$
11: $L_{KL} = KL(o_{i-1}||\mathcal{N}(0, 1))$
12: backpropagate $L_{Recons} + \lambda_{KL} * L_{KL}$
13: **return** $o_{i-1}$

---

states, it can operate cautiously with shorter sequences. However, this termination condition is not dependent on the state, so it cannot respond to sudden changes in the state.

### 3.4.3 TERMINATION BY UNEXPECTEDNESS

In this setting, we utilize the predictor network $P$ to decide the length of action sequences (Figure 3 (b)). The intuition is that the predictor network $P$ also acts as a world model (Ha & Schmidhuber, 2018), which models the environment dynamics. When things are within expectation, the decoder $D$ can go ahead and output longer sequence. When the agent encounters unfamiliar states, $D$ should become more cautious and output shorter sequence. This termination condition (referred to as Term-predict) depends on both the option and the state.

During training, we set a threshold $\delta$ on the prediction loss $L_{pred}$. When $L_{pred}$ is bigger than $\delta$, the encoder $D$ terminates the output, then the HRL policy will output a new option for subsequent actions. The training algorithm is detailed in Algorithm 1. The option learned in this way is supposed to be more robust to state changes. However, this method relies on a good predictor. The length of the sequence is dependent on the hyper parameter $\delta$.

### 3.5 POLICY GRADIENT OVER OPTIONS

Given the option, we employ Semi-MDP framework to train an HRL agent at a higher level. Because the option is in continuous space, we use policy gradient algorithm and derive the algorithm over options. Note that although our option is continuous, our temporal abstraction could be applied to discrete problems, as long as the RVAE outputs discrete actions.

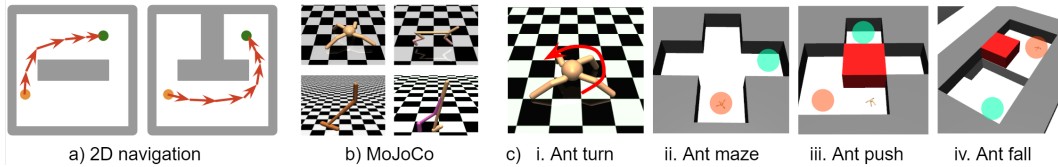

Figure 4: Evaluation tasks. a) 2D navigation task. b) MoJoCo tasks including Ant, HalfCheetah, Hopper and Walker2d. c) Ant tasks including AntTurn, AntMaze, AntPush and AntFall.

The gradient computation follows the same principle with the flat RL problem formulated by Sutton et al. (2000); Schulman et al. (2017). The gradient over the parameters has the form:

$$\nabla_\theta U(\theta) = \mathbb{E}_{(s_t,o_t) \sim \pi_\theta}[\nabla_\theta log \pi(o_t|s_t;\theta)A(s_t,o_t)] \tag{10}$$

where the $\pi(o|s;\theta)$ is the high-level policy planning in state-option space. The advantage function $A(s_t, o_t)$ is designed as the accumulative reward subtracting a baseline function:

$$A(s_t, o_t) = R(s_t, o_t) - V(s_t) \tag{11}$$

where $R(s_t, o_t)$ is the cumulative discounted reward from time $t$ to the future, and the baseline function $V(s)$ is updated with the TD algorithm (Sutton & Barto, 2018) (assuming the upcoming options are of length $t_1, t_2, ...$):

$$V(s_t) \leftarrow V(s_t) + \alpha(r_{t \to t+t_1} + \gamma^{t_1} V(s_{t+t_1}) - V(s_t)) \tag{12}$$

where $r_{t \to t+t_1}$ is the discounted cumulative reward during executing $o_t$ between $t$ and $t + t_1$:

$$r_{t \to t+t_1} = r_t + \gamma r_{t+1} + ... + \gamma^{t_1 - 1} r_{t+t_1 - 1} \tag{13}$$

In the experiments, we employ PPO algorithm (Schulman et al., 2017). In general, the policy gradient over option and the policy gradient over action are similar. Most of the existing RL algorithms such as TRPO, TNPG (Schulman et al., 2015) and SAC (Haarnoja et al., 2018b) are applicable.

## 4 EXPERIMENTS

We first consider a 2D navigation task for proof of the concept. Then we apply our temporal abstraction framework to robotic control tasks in MuJoCo (Todorov et al., 2012; Brockman et al., 2016). At last we will move to more challenging tasks, which are also used by Haarnoja et al. (2018a) and Nachum et al. (2018), and are hard to solve by non-hierarchical methods.

The experiments in this section are generally performed in three steps: 1) collecting the experiences using a flat PPO agent; 2) learning the option based on the TAIC algorithm; 3) training HRL based on the option. We implement a one-layer LSTM (Hochreiter & Schmidhuber, 1997) with 64 hidden units for both the $E$ and $D$, and three-layer MLPs for the $RL$, $F$ and $P$. We use learning rate 0.01 for the $E$, $D$ and $P$, learning rate 0.0003 for the $RL$, 0.001 for $F$. We also balance multiple losses with different weights: $\lambda_{kl}$ is used for the $L_{kl}$ in RVAE; $\lambda_{adv}$ is used for $L_{adv}$ for $F$.

### 4.1 2D NAVIGATION TASK

We first test our framework using a 2D navigation task, which is a toy example that allows us easily visualize the option we learned from the experience. As Figure 4 (a) shows, our environment is a 10m by 10m room with obstacles. Both the state and the action are continuous, represented by $(x, y)$ and $(v_x, v_y)$ respectively. The goal is to navigate from the start location (orange circle) to the goal location (green circle). The reward for reaching the goal is 100, and -1 otherwise.

We collect the experience using a flat PPO agent, running on multiple tasks with random start and goal location. Then we train an RVAE network using the collected action sequences. Figure 5 (a) shows the interpolation between two options. We first randomly sample two action sequences from

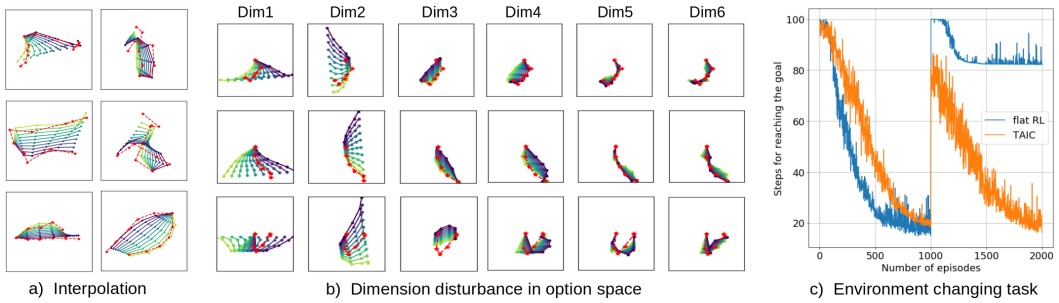

Figure 5: 2D navigation task. a) Interpolation in option space. b) Disturbing one dimension in the option space. c) Comparing the flat RL and TAIC on the ability of recovering from sudden environment change (average over 10 trials).

the testing set (shown in red dash lines), and encode them into two options using the network $E$. We do a linear interpolation between these two options and then decode those options into a set of action sequences using network $D$. The first thing we notice is that the RVAE nicely capturing the direction and curvature of the sequences and usually outputs a smoother version (solid lines) of the original sequence (red dash lines). Further, the interpolations between two sequences smoothly transfer from one to the other, denoting that we have a smooth option space.

In Figure 5 (b), we visualize how each dimension in the option space encodes different information. The red dash line shows an action sequence sampled from the testing set. This action sequence is encoded into an option which is a vector with 6 real values. We modify one dimension and decode the options into action sequences shown with the solid lines. We notice the first two dimensions control the ending point of the action sequence, and dimension 3 and 4 control the curvature, dimension 5 and 6 control more subtle properties. Note that not all the trials of training get the same results. Sometimes the properties are mixed, but Figure 5 illustrates one of the most common results.

Next, we evaluate the ability to recover from a sudden change in the environment. Sutton et al. (1999) and Bacon et al. (2017) also presented a similar experiment. Initially the environment is set up as Figure 4 (a) left. After 1000 episodes, an obstacle blocks the shortest path, and creates a trap (Figure 4 (a) right). We reset the standard deviation output of the PPO algorithm to encourage exploration. The flat RL is hard to recover from this situation since the exploration at each time step still gives a similar trajectory, which leads to the trap. While the exploration in the option space results in more diverse trajectories. We average 10 repeated experiments for both settings. As can be seen in Figure 5 (c), the TAIC recovers from the environment changes for all 10 trials. While the flat RL agent find the goal in 20% trials.

## 4.2 ROBOTIC CONTROL TASKS

In the robotic control domain, we utilize the MuJoCo tasks in the OpenAI Gym environment (Todorov et al., 2012; Brockman et al., 2016). The goal is to learn a control policy $\pi : \mathcal{S} \mapsto \mathcal{A}$, so that the robot runs forward as fast as possible (Figure 4 (b)). Both the state and the action are vectors of continuous values. We present two ways of evaluating the TAIC framework. Firstly, we qualitatively evaluate the learned option by visualize the option space. Secondly, we apply the learned option to the HRL training and compare the performance.

### 4.2.1 VISUALIZATION OF THE OPTION SPACE

Evaluating the high dimensional option space is difficult. We observe that the losses in the option learning process ($L_{rvae}$, $L_{pred}$ and $L_{adv}$) could not reveal the underlying structure of the option space. We propose a visualization method that provides a qualitative evaluation of the option space.

We now consider how to visualize the correlation between the option and state change. We randomly sample the options $\{o^{(0)}, o^{(1)}, o^{(2)}, ...\}$ from the option space, and decode the options into action sequences $\{a^{(0)}_{0...k_1}, a^{(1)}_{0...k_2}, a^{(2)}_{0...k_3}, ...\}$. All the action sequences are applied to the same start state,

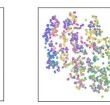 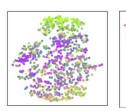 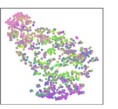 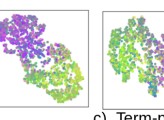 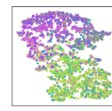

a) Fix-len (no constraint)     b) Term-output     c) Term-predict

Figure 6: Visualization of option space w.r.t. state change. a) Option obtained by RVAE wo/ constraints. b) Option w/ constraints and Term-output. c) Option w/ constraints and Term-predict. The points in (c) are most ordered w.r.t. the color. This suggests the option distribution in (c) is most correlated with state change.

obtaining a set of end states $\{s^{(0)}, s^{(1)}, s^{(2)}, ...\}$. Since the start states are the same, we use the end state to represent the state changes.

Figure 6 shows the visualization of the option space trained for HalfCheetah-v1 task. Each point in the figure represents an option-state pair $(o^{(i)}, s^{(i)})$. The options and the end states are first converted to 2D vectors using t-SNE (Maaten & Hinton, 2008), respectively. The 2D vector of the end states are associated to the location of the points in the figure, while the color of the points represents the distribution of the options. This is to say, if the option distribution is more coupled with the state change, the points will be arranged better with respect to the color. As we can see from Figure 6, with information-theoretic constraints the options and state changes become more correlated.

### 4.2.2 HRL USING OPTIONS

We evaluate the option on four benchmark tasks in simulation. Figure 7 compares the HRL performance with different termination conditions. We observe the HRL using TAIC outperform the flat PPO in 3 out of 4 tasks. In the HarfCheetah and the Ant task, the curve of TAIC starts higher (because the random policy over the option space already gets positive rewards) and converges faster. One thing should be notice is that the HRL policy is updated in a much lower frequency than the flat RL, because the options are executed in multiple time steps (a typical average length is 5 time steps).

In the Walker2d task, our method does not outperform the flat RL. Our hypothesis is that the agent is more unstable in this task. In our current setup, the decoder $D$ does not depend on the state $S$. It acts like an open-loop controller that is more vulnerable to unstable and fast-changing states. However, our option model can be extended to close-loop sub-policies $\pi(a|o, s)$ that also depends on the state. We will explore this idea in the future study.

The different termination conditions have pro and cons. The Fix-len termination condition is more simple and stable in most of the cases. The ones with Term-output and Term-predict termination condition get better results in some cases, but they require more tuning of the parameters such as the predict threshold, which controls the average length of the sequences.

The experiences used for training the option model is a critical factor that decides what the agent can learn (as humans can also be biased by their previous experiences). We tested two ways of selecting the experiences: random selecting the experiences and selecting experiences with higher rewards. From our experiments, the agent learns faster and performs better in the latter setting.

### 4.3 TRANSFER THE OPTION TO MORE DIFFICULT TASKS

The TAIC framework provides an efficient way to transfer the past experiences to unseen and more sophisticated tasks. We apply the options learned from the above MoJoCo Ant-v1 running task, to novel tasks shown in Figure 4 (c). In the Ant turn task, the agent is awarded by making left turn as fast as possible. The goal of the other three tasks (Ant maze, Ant push and Ant fall) is to move from the start location (orange circle) to the goal location (green circle). The reward is the negative of the distance between the agent and the goal. The last two tasks require the agent to manipulate the red box in a right way, in order to avoid being stuck in sub-optimal trap states.

As shown in Figure 8, TAIC (green) achieves higher rewards and quickly converges to the goal state in the first three tasks. In contrast, the flat RL (blue) method is not able to solve these tasks in 3

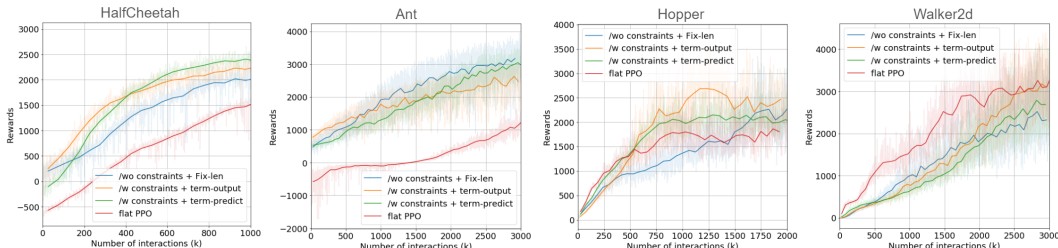

Figure 7: Evaluating the performance of TAIC by HRL training on MuJoCo tasks. The proposed framework with different termination conditions are compared with the flat RL. Each case is an average over 5 repeated training.

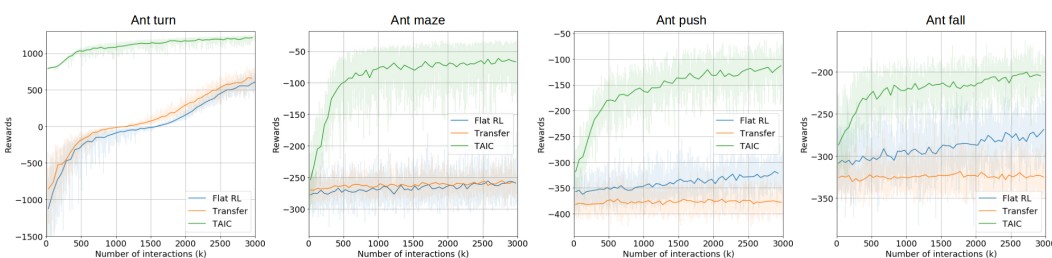

Figure 8: Transfer capability. We compare the proposed TAIC framework with two baseline methods on more challenging tasks. The TAIC framework efficiently transfers the options learned in simpler Ant-v1 task to these novel and more complex tasks.

million interactions without any task transfer treatment. We also apply transfer learning on the flat RL (orange) by using the Ant-v1 policy as an initialization. As expected, transfer learning does not bring comparable benefit to the flat RL, indicating sharing weights between different tasks is less efficient. The last two experiments also show that the transferred policy (orange) may even degrade the performance. This does not happen to our TAIC framework since it transfers high level abstracted knowledge in the form of options.

## 5 CONCLUSION

This paper presented a general HRL framework TAIC for learning temporal abstraction from action sequences. We formulate the temporal abstraction problem as learning latent representations (called options) over action sequences. In order to learn a better representation, we derive theoretically on how to regularize the option space and give an applicable solution of adding constraints to option space. In the experiments, we try to reveal the underlying structure of the option space by visualizing the correlation between options and state changes. We showed qualitatively and quantitatively that our options encode meaningful information and benefit the RL training. Furthermore, the TAIC framework provides an efficient tool to transfer the knowledge learned from one task to another. Our framework can be applied together with all kinds of RL optimization algorithms, and can be applied to both discrete and continuous problems.

This work brings many new directions for future studies. As we currently learn the RL task and the option separately, the option could not be improved with the improvement of the policy. In theory, it is entirely feasible to jointly optimize the two parts, or at least train them alternately. As mentioned above, the current sub-policy acts like an open-loop controller. So learning a close-loop sub-policy beyond the RNN decoder will be one of the focus areas of our future studies. We would also like to apply the TAIC framework to discrete problems and with other RL algorithms such as DQN and SAC. This could bring more insights to further improve the framework.

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
