# OpenReview forum: "Learning Temporal Abstraction with Information-theoretic Constraints for Hierarchical Reinforcement Learning"
_ICLR.cc/2020/Conference — Reject_

### Official Review · AnonReviewer2 · 2019-10-20
**Official Blind Review #2**

**Rating:** 3

**Review:**

The authors propose a Hierarchical Reinforcement Learning (HRL) framework based on learning latent representations of action sequences. They use a Recurrent Variational Autoencoder (RVAE) to encode action sequences from previous experience or expert demonstration. They regularize representations using the fact that these representations should contain information about state changes, but not the states themselves.

The approach is developed both intuitively and theoretically. Detailed visualisations demonstrate that the results match the intuition. The paper is well written and relatively easy to follow. The related work section is wanting - see below.

Comments

If we understood correctly, E, D, F, and P are pre-trained in an unsupervised way from expert demonstration as in imitation learning. We ask the authors to clarify this in the paper.

In Algorithm 1, we don't see how F is trained. Is this missing or not part of the algorithm at all? Also, in line 10, how is MSE calculated if i != j?

In the experimental section, experience is collected using a PPO agent. A flat policy is used as a baseline. Is the experience collection included in the number of interactions or just used to pre-train (parts of) the model? In the latter case, the comparison might be improper.

Also, flat policy might be a weak baseline given recent progress on HRL. Comparison with other recent methods such as those in [1][2][3] would be desirable, but not a must.

Typos etc

Page 3, Section 3.3, instead of "however" I suggest "on the other hand" or similar.
Page 4, Section 3.3, "summation of two conditional entropies" instead of "two conditional entropy".
Page 9, Section 4.2.2, "noticed" instead of "notice".

Related work

We don't think this is the first time an RVAE has been used for encoding action sequences. SeCTAR [1] also uses an RVAE to encode trajectories (both states and actions) for HRL. The authors should include a reference to the paper and discuss similarities and differences between SeCTAR and their own work.

Other missing recent related works include HIRO [2] and Hierarchical Actor Critic [3].

They write: "the HRL often requires explicitly specifying task structures or sub-goals (Barto & Mahadevan,2003; Arulkumaran et al., 2017). How to learn those task structures or temporal abstractions automatically is still an active studying area." "Some early studies try to find sub-goals or critical states based on statistic methods (Hengst, 2002; Jonsson, 2006; Kheradmandian & Rahmati, 2009). More recent work seeks to learn the temporal abstraction with deep learning (Florensa et al., 2017; Tessler et al., 2017; Haarnoja et al., 2018a). However, many of these methods still require a predefined hierarchical policy structure (e.g. the number of sub-policies), or need some degree of task-specific knowledge (e.g. hand-crafted reward function)."

These are rather recent references. To our knowledge, however, the first HRL with temporal abstraction was published 1990-1991. See the references in section 10 of the overview http://people.idsia.ch/~juergen/deep-learning-miraculous-year-1990-1991.html  "Hierarchical RL (HRL) with end-to-end differentiable NN-based subgoal generators [HRL0], also with recurrent NNs that learn to generate sequences of subgoals [HRL1] [HRL2]. An RL machine gets extra inputs of the form (start, goal). An evaluator NN learns to predict the rewards/costs of going from start to goal. An (R)NN-based subgoal generator also sees (start, goal), and uses (copies of) the evaluator NN to learn by gradient descent a sequence of cost-minimising intermediate subgoals. The RL machine tries to use such subgoal sequences to achieve final goals." See also [HRL4] on another way of discovering appropriate subgoals. How does the work of the authors go beyond this original work on learning temporal abstractions for HRL?


Additional References mentioned above:

[1] John Co-Reyes, Yu Xuan Liu, Abhishek Gupta, Benjamin Eysenbach, Pieter Abbeel, and Sergey Levine. Self-Consistent Trajectory Autoencoder: Hierarchical Reinforcement Learning with Trajectory Embeddings. ICML 2018.
[2] Ofir Nachum, Shixiang Gu, Honglak Lee, and Sergey Levine. Data-Efficient Hierarchical Reinforcement Learning. NeurIPS 2018.
[3] Andrew Levy, George Konidaris, Robert Platt, Kate Saenko. Learning Multi-Level Hierarchies with Hindsight. ICLR 2019.

Overall, we believe this is a promising paper, but we are not sure if it is ripe for publication at ICLR in its current state. For now, we'd lean towards rejecting this submission, but we might change our minds, provided the comments above were addressed in a satisfactory way. Let us wait for the rebuttal.

**Experience Assessment:**

I have published in this field for several years.

**Review Assessment: Checking Correctness Of Derivations And Theory:**

I assessed the sensibility of the derivations and theory.

**Review Assessment: Checking Correctness Of Experiments:**

I assessed the sensibility of the experiments.

**Review Assessment: Thoroughness In Paper Reading:**

I read the paper thoroughly.

---

> ### Author Response · Authors · 2019-11-11
> **Response to Reviewer #2 (2/2)**
>
> >> Q4: We don't think this is the first time an RVAE has been used for encoding action sequences. SeCTAR [1] also uses an RVAE to encode trajectories (both states and actions) for HRL. The authors should include a reference to the paper and discuss similarities and differences between SeCTAR and their own work. Other missing recent related works include HIRO [2] and Hierarchical Actor Critic [3].
>
> A4: Thanks for pointing out these related works. We’ll include these references in the paper.
>
> The SeCTAR algorithm is very interesting and related. Both of us proposed a HRL framework by learning a latent representation from trajectories using RVAE. Both works learned a predictive model: SeCTAR learned a model capturing environment dynamics, while we use a predictive model to regularize the RVAE.
>
> However, there are significant differences. SeCTAR learns a latent representation from state sequences, while our proposed TAIC learns from action sequences. From our understanding, this difference comes from the different motivations and intuitions behind the two frameworks. SeCTAR focuses on learning a sub-policy and predictive model that follow a state trajectory. The intuition is that instead of learning a fine-grain temporal predictive model, SeCTAR only needs to predict the temporally extended behaviors of the sub-policy. The learned model and sub-policy can facilitate a higher-level model-based method such as MPC. On the other hand, our intuition (as we described in the reply to reviewer #1) is that there are USEFUL PATTERNS in action sequences. Take human motion for example, you can clearly distinguish raising hands by a normal person and a person with Parkinson disease. Although there are infinite ways of raising hands, a lot of action combinations (e.g. the way a person with Parkinson disease raising his hand) are not that useful and common. Learning the patterns of useful action sequences could help us better control the body and achieve our goals more efficiently. So our focus is on learning useful action representations.
>
> SeCTAR has some nice properties such as close-loop sub-policy, exploration module and an online iterative learning mechanism. Incorporating these ideas into our TAIC framework would be very interesting future work, as we have discussed in the last section. We conjecture one disadvantage of the SeCTAR is that it could be more sensitive to environmental changes, since the sub-policy is tightly coupled with the environment dynamics. While the latent representation of TAIC purely depends on the actions, the learned skills could be easily transferred between different tasks (without any finetuning of the sub-policy), as we shown in Fig 8. In addition, SeCTAR use fixed-length trajectory, while we have investigated several termination conditions, which allow the sub-policy outputs variable length trajectories.
>
> >> Q5: To our knowledge, however, the first HRL with temporal abstraction was published 1990-1991. See the references in section 10 of the overview http://people.idsia.ch/~juergen/deep-learning-miraculous-year-1990-1991.html. How does the work of the authors go beyond this original work on learning temporal abstractions for HRL?
>
> A5: Thanks for pointing out those classic papers. Personally speaking, I have been greatly inspired by the work of Prof Schmidhuber and his team. I think a lot of their ideas (such as recurrent world models, curiosity, data compression, etc.) are so beyond their times, and could be dug deeper in the current context. Those early work presented innovative methods and exciting results, which inspired numerous following research. We are just one of them that trying to push the original idea towards more scalable, generalizable and applicable direction.

---

> ### Author Response · Authors · 2019-11-11
> **Response to Reviewer #2 (1/2)**
>
> >> Q1: If we understood correctly, E, D, F, and P are pre-trained in an unsupervised way from expert demonstration as in imitation learning. We ask the authors to clarify this in the paper.
>
> A1: Yes. $E$, $D$, $F$ and $P$ are trained in the first phase using action-state sequences, which in our experiments come from the agent’s own experiences. They can also come from expert demonstration, in which case we speculate will be better. We haven’t done much study on how the experience could influence the learned option. But some primitive experiments show that better experiences result in better options, as briefly discussed in the 4th paragraph of 4.2.2.
>
> >> Q2: In Algorithm 1, we don't see how F is trained. Is this missing or not part of the algorithm at all? Also, in line 10, how is MSE calculated if i != j?
>
> A2: We omitted $F$ in the Algo 1 because it is not part of termination condition. But $F$ is trained together with $E$, $D$ and $P$ as described in Section 3.3. The network $F$ takes in the option and outputs the start state $s$ and the end state $s’$. The weights of $F$ is optimized to minimize the prediction loss $L_{adv}$. At the same time, the weights of $E$ are regularized by maximizing the loss $L_{adv}$.
>
> Thanks for pointing out the mistakes in the Algo 1. We set $j=i-1$. The reconstruction loss is calculated by $L_{Recons}=MSE(\hat{a}_{0…(i-1)}, a_{0…(i-1)})$. We’ll update the paper accordingly.
>
> >> Q3: In the experimental section, experience is collected using a PPO agent. A flat policy is used as a baseline. Is the experience collection included in the number of interactions or just used to pre-train (parts of) the model? In the latter case, the comparison might be improper.
> Also, flat policy might be a weak baseline given recent progress on HRL. Comparison with other recent methods such as those in [1][2][3] would be desirable, but not a must.
>
> A3: The interaction in the experience collection is not included. We agree that the comparison in this setup advantages the HRL algorithm, because the HRL method is exposed to extra information, which is the past experiences. The main message we’d like to convey is that our framework presents an efficient way of taking advantage of the past experiences to accelerate the learning task. We didn’t count the interaction in the pre-train process, because the experiences can also come from expert demonstration or from the pre-training on simpler tasks (e.g. in the setup of Section 4.3).
>
> We agree that adding more baselines on HRL side would enhance the paper. However, in this work, we propose to handle HRL with a new framework. We have a two-stage procedure, which is different from existing methods. Our primary intention is to show that performance of an RL learner could be improved based on our proposed framework. One can improve the performance by inserting a more powerful RL algorithm, or learning options from better experiences (using expert demonstration for example). By showing that our framework can learn useful abstractions from action sequences, and showing that the learned option could accelerate training of both seen and unseen tasks, we demonstrate the framework is effective and can be improved in many ways, as we discussed in the last section. In our opinion, the experiments have shown strong evidence of the effectiveness of our framework.

---

### Official Review · AnonReviewer1 · 2019-10-22
**Official Blind Review #1**

**Rating:** 1

**Review:**

Summary:

This paper develops a method for learning a latent action representation based on prior experiences (and specifically, prior action sequences). Additionally, the paper proposes to regularize the learning of this representation using an information-theoretic constraint, yielding Temporal Abstraction with Information-theoretic Constraints (TIAC). Indeed, one promise of HRL is to allow for learning and decision making algorithms to take the long-term consequences of a decision into account when planning, exploring, assigning credit, or simply acting. The options framework (Sutton, Precup, and Singh; 1999) is a promising and well-studied toolkit for investigating these capacities of HRL. For this reason, the topic of the paper is well chosen: continuing to understand how options can benefit and accelerate RL in rich environments is an important direction for research. The idea at the core of the paper is new to my knowledge: learning an encoding of action sequences with a continuous latent representation. It could be a promising technique for HRL. Experiments are conducted to evaluate the effectiveness of the method in several environments, including a continuous gridworld, control tasks, and problems involving transfer learning.

Verdict: Due to lack of clarity in describing the main methods, and missing comparison to any HRL/option baselines, I recommend rejection.

More Detail:

The paper is lacking clarity in its current form. I view the main contribution as the development of the architecture and loss function that together learn an appropriate latent action representation. There are two key issues with clarity at present: 1) The presentation of the core technical contributions could be improved (see comments below in "Q1"), and 2) Motivation for this style of option learning is missing, with evidence that the proposed method is in fact learning an appropriate thing.

Toward (1): I provide suggestions where clarity could be improved below in "Q1'.

Toward (2): There are a few aspects of the motivation that could be improved. First, the paper mentions that the learned options/representation will help in planning, but planning is not studied in the paper. For example: "Further, the interpolations between two sequences smoothly transfer from one to the other, which is a desired property to have during planning, because the smooth option space provides the RL algorithm with a better search space." By my reading of the paper, this claim is not studied. Similarly, in the intro: "...allow us to do planning at a higher level, and easily transfer the knowledge between different tasks". Including experiments that explicitly evaluate the capacity of the learned representation to carry out planning would help support these claims. Or, alternatively, the contribution could be focused to model-free and policy-based learning, which is where the empirical evidence currently offers the most support. Second, no HRL baselines are compared to in the experiments. One natural comparison to include would be to the Option-Critic, which was the first technique for combining option learning with deep RL. To determine whether TIAC is a sensible approach to learning and using options, a comparison to at least one other option learning method is needed. The paper currently highlights the fact that the option-critic requires a pre-specified number of options: this is true, but it is not discussed why is this problematic, or how the current proposal remedies this difficulty. Others that may be relevant include FuN (Vezhnevets et al. 2017), the recent methods of Nachum et al. (2018), among others (Tiwari and Thomas 2019, Harb et al. 2018, Harutyunyan et al. 2019, Levy et al. 2019).

In short: the results here are promising, so I encourage the authors continue in this direction. The paper will be improved if the presentation of Section 3 is sharpened (see questions regarding clarity below) and a comparison with relevant baselines is included.

Main Questions:

Q1: The exposition of the main method (Section 3) was unclear to me. Here are a few questions I was left with:

	(a) Why is the posterior (on $o$) conditioned only on the action history, and not state?
	(b) Additionally $o$ is being treated as a random variable through 3.2. So, what is $o$? Where is the randomness coming from?
	(c) Section 3.3 states "it encodes the action sequences with respect to the L2 distance in action space". Does this mean the action space is always a subset of $\mathbb{R}$? But, it looks like $\mathcal{A}$ is just defined as some set: in Section 3.1, "$\mathcal{A}$ is the action set". So, I am confused as to what the $L_2$ is distance defined with respect to. If the actions are always assumed to be real numbers that is entirely okay, but it would be helpful to have that stated early on. From the additional text in Section 3.3, it sounds like the transition function of each action is involved in computing this distance ("...only have small difference in each step of action. Due to the error compounding, the two sequences...").
	(d) How is the estimate of the posterior actually used to act? The output of "D" in Figure 2 is $\hat{a_{0...k}}$. What is the type of this entity? Is it guaranteed to be an element of $\mathcal{A}$? If so, then the "option" here is a policy that maps $o$ and the action history $a_{0...k}$ to a new action, correct? Ah, so in Figure 3, it looks like D will have different output depending on how the termination condition is handled. Are the actions output by $D$ then executed by the RL agent, or is there some additional decision making that goes on downstream?
	(e) Early on the section states "In contrast to precisely reconstructing the action sequences, our goal is to extract the latent variable capturing  the information which could benefit RL training." It might be helpful to include some intuition about what this information would look like. It's unclear why action history would be all that meaningful on its own (without say, the state history). It would help the section to provide some intuition for such a latent variable existing; is there an idealized, simple case that would help convey the idea? Note that this proposal comes across as different from the original proposal of the options framework: As an example, Sutton, Precup, and Singh (1999) say: "options enable temporally abstract knowledge and action to be included in the reinforcement learning framework in a natural and general way". This temporally abstract knowledge need not be a function of the entire action history. I like this aspect of the method as it makes the proposed algorithm quite novel, but the motivation for why this should work didn't come through for me.
	(h) Should the mutual information in Eq. 4 be the conditional mutual information given $a_{0...k}$? (Same question for the remaining uses of $I$ and $H$).
	(i) It is unclear how the option learning coordinates with the RL algorithm used. That is, suppose we train the HRL component to learn the mapping from $s, a_{0...k}$ to the constituents identified in Figure 2/3. Where does the actual RL take place? Does the algorithm just execute the actions output by $D$ at each time step?

Q2: In the first experiment, it is stated: "because the smooth option space provides the RL algorithm with a better search space." Any thoughts as to why this is true? Including some discussion here might help motivate the approach.


Minor Comments:

	C1: I do not understand Figure 6. The color is said to denote "the distribution of options", but I couldn't quite make out what this was, precisely. It would be helpful to know the range of values it can take on, and how those values map to the displayed colors. Moreover, what is the take away from the figure? The text states "with information-theoretic constraints the options and state changes become more correlated" but I am having trouble connecting that claim with the visuals themselves. Some additional discussion here would be really helpful.

	C2: In Figure 5, what does "dimension disturbance in option space" mean?

Minor Typos/Writing Suggestions [did not affect evaluation]:
	Abstract:
	- "Applying reinforcement learning (RL) to"::"Applying reinforcement learning (RL) algorithms to"
	- I am having trouble parsing this phrase: "to learn new tasks on higher level more efficiently". Perhaps: "to learn new tasks at a higher level of abstraction more efficiently"
	- "over benchmark learning problems"::"over baseline learning algorithms on benchmark problems"

	Sec. 1 (Intro):
	- Plural acronyms tend to have an 's' at the end. So: Recurrent Variational AutoEncoders (RVAEs).
	- "conveys meaningful information and benefit the RL training"::"conveys meaningful information and can benefit learning"

	Sec. 2 (Related Work):
	- "the policy sketches"::"policy sketches"
	- Personal preference, by I always prefer "use" to "utilize".

	Sec. 3 (Approach):
	- Your $\mapsto$ operators should be replaced by $\rightarrow$. The $\mapsto$ operator indicates what is applied to elements on the left, while $\rightarrow$ specifies the domain and codomain of the function. Thus, the $\mapsto$ variation would be $P : (s,a) \mapsto s'$. The story is the same for $\beta$: it should read "$\beta : \mathcal{S} \rightarrow [0,1]$". Note that this (using $\rightarrow$) is how Sutton, Precup, and Singh (1999) define $\beta$ as well.
	- "Sub-policy is defined as a function over the random variable."::"Now, the sub-policy is defined as a function over the random variable."
	- Not a sentence: "So that the options with similar consequences become closer in the option space." Consider combining with the previous sentence.
	- This sentence runs on: "Given a set of past experiences...". Consider defining $\Lambda$ first as its own sentence, then definines the problem. Something like: "We let $\Lambda = ...$ Then, our problem is to learn...".
	- "it is empirically shown"::"it has been demonstrated empirically"
	- Latex quote issue: "”go reach the door".
	- In Equations 4-9: in general, mutual information is a function of random variables. Is $o$ a random variable? For instance I have trouble expanding $H(o)$. What is $p(o)$?
	- "the encode $E$ is regularized"::"the encoder $E$ is regularized"

	Sec. 4 (Experiments):
	- "task for proof of the concept"::"task as a proof of concept"
	- "that allows us easily visualize the option we learned from the experience"::"that allows us to easily visualize the options learned from experience"
	- "that the RVAE nicely capturing the direction":::"that the RVAE captures the direction"
	- Misuse of $\mapsto$: "learn a control policy $\pi : \mathcal{S} \mapsto \mathcal{A}$" should be "learn a control policy $\pi : \mathcal{S} \rightarrow \mathcal{A}$" or "learn a control policy $\pi : s \mapsto a$".
	- "HarfCheetah"::"HalfCheetah"

References:

Vezhnevets, Alexander Sasha, et al. "Feudal networks for hierarchical reinforcement learning." Proceedings of the 34th International Conference on Machine Learning-Volume 70. JMLR. org, 2017.

Nachum, Ofir, et al. "Data-efficient hierarchical reinforcement learning." Advances in Neural Information Processing Systems. 2018.

Tiwari, Saket, and Philip S. Thomas. "Natural option critic." Proceedings of the AAAI Conference on Artificial Intelligence. Vol. 33. 2019.

Harb, Jean, et al. "When waiting is not an option: Learning options with a deliberation cost." Thirty-Second AAAI Conference on Artificial Intelligence. 2018.

Harutyunyan, Anna, et al. "The Termination Critic." AISTATS 2019

Levy, Andrew, et al. "Learning multi-level hierarchies with hindsight." ICLR 2019.

**Experience Assessment:**

I have published in this field for several years.

**Review Assessment: Checking Correctness Of Derivations And Theory:**

I carefully checked the derivations and theory.

**Review Assessment: Checking Correctness Of Experiments:**

I assessed the sensibility of the experiments.

**Review Assessment: Thoroughness In Paper Reading:**

I read the paper thoroughly.

---

> ### Author Response · Authors · 2019-11-11
> **Response to Reviewer #1 (3/3)**
>
> >> Q10: Should the mutual information in Eq. 4 be the conditional mutual information given $a_{0...k}$?
>
> A10: Yes. The $o$ is conditioned on $a_{0...k}$. We omit the $a_{0...k}$ for simplification.
>
> >> Q11: It is unclear how the option learning coordinates with the RL algorithm used. That is, suppose we train the HRL component to learn the mapping from $s, a_{0...k}$ to the constituents identified in Figure 2/3. Where does the actual RL take place? Does the algorithm just execute the actions output by $D$ at each time step?
>
> A11: Yes, you are correct. There are two learning phases. First we do the option learning, which involves every component except the RL in Figure 2. In the second phase, we train RL using PPO (it can be any RL algorithms) on the specific task. The policy pi trained by PPO algorithm outputs an option $o$, which is decoded to a sequence of actions $a_{0...k}$, which will be executed in the environment.
>
>
> >> Q12: I do not understand Figure 6. It would be helpful to know the range of values it can take on, and how those values map to the displayed colors. Moreover, what is the take away from the figure?
>
> A12: As described in Section 3.3, we apply constraints on the encoder, so that the option is more correlated with state changes. Fig 6 gives a qualitative evaluation on this. How do we achieve this? The short answer is we execute options and see how the state changes. We first randomly sample 1000 options $\{o_0, … o_{999}\}$ (specific for the 2D navigation task, each option is a 6-d vector). Then we decode these options into action sequences $\{a_{0...k_0}, a_{0...k_1}, ... , a_{0...k_{999}}\}$. We apply these action sequences to the same state $s_{start}$, resulting in 1000 different end states $\{s_0, ... , s_{999}\}$. Thus, we visualize the correlation between 1000 options and 1000 end states (because the start states are the same, we use end state to denote state changes). How do we visualize the correlations? We now have 1000 one-to-one correspondence between option and state changes. The option is a 6-d vector, while the state is a 2-d vector. We use t-SNE to convert option to 3-d vector, which is associated to an RGB color. The state 2-d vector is associated to the 2-d coordinates. The take away message is that each point in the figure denotes a option-state correspondence. The more ordered (e.g. Figure 6, c) of the color, the more correlated of the state-option pair.
>
> >> Q13: In Figure 5, what does "dimension disturbance in option space" mean?
>
> A13: Each option is a 6-d vector. Fig 5 (b) visualizes what each dimension is encoding. How do we achieve this? The short answer is we disturb one dimension of the option at each time, decode the changed option and see how the action sequence change. We first encode an action sequence $a_{0...k}$ to an option $o$, disturb one dimension of $o$ by changing it gradually from -2.0 to 2.0 while fixing the other five dimensions. Then we decode the changed option $o’$ into action sequence. In Fig 5 (b), the red trajectory is the original action sequence $a_{0...k}$, the other trajectories are the disturbed ones. We find each dimension of the option is interpretable. The first two dimensions are encoding the moving direction of the action sequence, while the rest are encoding the curvature.

---

> ### Author Response · Authors · 2019-11-11
> **Response to Reviewer #1 (2/3)**
>
> >> Q7: Section 3.3 states "it encodes the action sequences with respect to the L2 distance in action space". I am confused as to what the is L2 distance defined with respect to.
>
> A7: Sorry for the confusion. We followed the work of VAE (Kingma & Welling (2013)) and RVAE (Ha & Eck (2017)).
>
> First, let me clarify the notation. The $\mathcal{A}$ denotes the action set. We use lower-case $a$ to denote a single action, which is a real value vector $\mathbb{R}^n$ in our case. The $a_{0...k}$ denotes a sequence of actions.
>
> In Section 3.2, we describe how we use the RVAE to encode the action sequences $a_{0...k}$  into an option $o$ (Ha & Eck (2017) did similar things to encode a sequence of strokes) (in which case an option $o$ is a real value vector). The RVAE minimizes a reconstruction loss plus a KL divergence loss (Eq 3). The reconstruction loss is defined as the L2 distance between two action sequences.
>
> In Figure 1, we use the arrow to denote action, and the arrow trajectory denotes the action sequences. In Figure 1, the purple and blue action sequences are closer, however, they have different consequences (the ending state). While the purple and green ones are more different, but they have the same consequence. Under the definition of VAE’s loss function, the blue and purple ones will be encoded closer. However, we argue it makes more sense to encode action sequences according to their consequences, which mean the purple and green one should be closer. This is the motivation of adding the information-theoretic constraints.
> Our method introduces a new regularization over the learned latent representation $o$ (or option $o$) in addition to the L2 distance. The new regularization encourages $o$ to encode action sequences in a way that options with similar consequences (state change) will be closer to each other.
>
> >> Q8: How is the estimate of the posterior actually used to act? The output of $D$ in Figure 2 is $\hat{a}_{0...k}$. What is the type of this entity? Is it guaranteed to be an element of $\mathcal{A}$? If so, then the "option" here is a policy that maps $o$ and the action history $a_{0...k}$ to a new action, correct? Ah, so in Figure 3, it looks like $D$ will have different output depending on how the termination condition is handled. Are the actions output by $D$ then executed by the RL agent, or is there some additional decision making that goes on downstream?
>
> A8: Sorry for the confusion. We find the Figure 2 is somehow too abstracted and confusing. In the above answer, we explained that we use RVAE to encode and decode action sequences and options. The RVAE has two parts, an encoder $E$ (maps action sequence $a_{0...k}$ to option $o$) and a decoder $D$ (maps opposite way). So the “option” is a latent representation of an action sequence. We first train this RVAE on pre-collected experiences (state-action sequences). Then the $D$ is fixed and works as a sub-policy in the HRL setting. We train a policy network that output option $o$, in order to maximize the reward, given a certain task. The option $o$ is decoded into a sequence of action by $D$, and executed in the simulation environment.
>
> >> Q9: Early on the section states "In contrast to precisely reconstructing the action sequences, our goal is to extract the latent variable capturing  the information which could benefit RL training." It might be helpful to include some intuition about what this information would look like. It's unclear why action history would be all that meaningful on its own (without say, the state history). It would help the section to provide some intuition for such a latent variable existing; is there an idealized, simple case that would help convey the idea?
>
> A9: Generally speaking, the intuition behind learning abstraction from action sequence is that there are USEFUL PATTERNS in action sequences. Take human motion for example, you can clearly distinguish raising hands by a normal person and a person with Parkinson disease. Although there are infinite ways of raising hands, a lot of action combinations (e.g. the way a person with Parkinson disease raising his hand) are not that useful and common. Learning the patterns of useful action sequences could help us better control the body and achieve our goals more efficiently. In section 3.3, we describe this objective by saying “extract the latent variable capturing the information which could benefit RL training”.

---

> > ### Comment · AnonReviewer1 · 2019-11-15
> > **R1 Response to (2/3)**
> >
> > Re A8: That makes sense, thank you for clarifying! One detail that is still unclear; when you say "we train a policy network that outputs option $o$", what is the input to that policy network? Judging by the figure, the input is a single state (rather than a sequence), correct? Does this mean that a single time step is passed into this network to generate $o$, which then gets decoded to $\hat{a}_{0, \ldots,k}$, which are potentially used to act for up to $k+1$ time steps depending on termination? If so this is quite different from the typical form of options which prescribes a policy that gets to react to the future states visited. It is okay if the two methods are different of course. Perhaps this is where some of the confusion was coming from. It is a point that may want to be emphasized.
> >
> > Re A9: Thank you for the response! In a similar spirit to my comment Re A8, options are typically used to capture action-correlations conditioned on a particular state or given a sequence of states visited by the option policy (see: Konidaris and Barto IJCAI 2007, for instance). My comment was pointing out that action correlations alone, without any amount of state history, are probably less useful. I take your reaction to suggest that action correlations alone can be useful. Regardless, my question here was clarified, so thank you!
> >
> > References:
> >
> > Konidaris, George, and Andrew G. Barto. "Building Portable Options: Skill Transfer in Reinforcement Learning." IJCAI. Vol. 7. 2007.

---

> > > ### Author Response · Authors · 2019-11-15
> > > **Author Response to Reviewer #1's Response (2/3)**
> > >
> > > Re: Re A8/A9: What you described in your comments are correct. Instead of a close-loop option proposed by the original work of Sutton et al, the option in our current setup is open-loop. It's more like a trajectory library in the robot planning field. We have a brief discussion in the experiment section and the conclusion section, designing and implementing a close-loop sub policy based on the proposed option would be our future work.

---

> ### Author Response · Authors · 2019-11-11
> **Response to Reviewer #1 (1/3)**
>
> >> Q1: (1) First, the paper mentions that the learned options/representation will help in planning, but planning is not studied in the paper. (2) "...allow us to do planning at a higher level, and easily transfer the knowledge between different tasks". Including experiments that explicitly evaluate the capacity of the learned representation to carry out planning would help support these claims.  (3) the contribution could be focused to model-free and policy-based learning, which is where the empirical evidence currently offers the most support.
>
> A1: We believe these questions are caused by our misuse of the word “planning”.
> (1) By planning, we meant to say the “HRL process” (i.e. the decision making process based on the learned options). We shouldn’t have used the word ‘planning’, because the process does not involve a model.
> (2) By “do planning at a higher level” we mean that we do HRL using the learned option. By saying “transfer the knowledge”, we use the Ant-Maze task, in which we show the option learned in a simpler task (Ant running) can benefit the learning of more difficult Ant-Maze tasks.
> (3) We believe the confusion comes from the misuse of the word “planning”. Thanks for the valuable suggestions.
>
> >> Q2: "Further, the interpolations between two sequences smoothly transfer from one to the other, which is a desired property to have during planning, because the smooth option space provides the RL algorithm with a better search space." By my reading of the paper, this claim is not studied.
>
> A2: This claim is just an intuitive explanation about why the learned option could benefit RL training. We show in gym tasks (Fig 7) and ant-maze tasks (Fig 8) that using option significantly speeds up the high-level learning. If we ask why using option is helpful, it’s very difficult to analyze. We try to show some qualitative analysis by visualizing the option space of the simple 2D navigation example (Fig 5). Figure 5 shows interesting results that you can do interpolation between two options, and then decode them into action sequences. Those action sequences form a smooth transition (Fig 5 (a)). This is not surprising, it’s kind of a known property of VAE. Intuitively speaking, search in a smooth option space should be easier to find a better solution than searching in an unsmooth one.
>
> >> Q3: No HRL baselines are compared to in the experiments.
>
> A3: In this work, we propose to handle HRL with a new framework. We have a two-stage procedure, which is different from existing methods (and which makes it harder to compare). Our primary intention is to show that performance of an RL learner could be improved based on our proposed framework. One can improve the performance by inserting a more powerful RL algorithm, or learning options from better experiences (using expert demonstration for example). By showing that our framework can learn useful abstractions from action sequences, and showing that the learned option could accelerate training of both seen and unseen tasks, we demonstrate the framework is effective and can be improved in many ways, as we discussed in the last section. In our opinion, the experiments have shown strong evidence of the effectiveness of our framework.
>
> >> Q4: The paper currently highlights the fact that the option-critic requires a pre-specified number of options: this is true, but it is not discussed why is this problematic, or how the current proposal remedies this difficulty.
>
> A4: The number of options has to be tuned for different tasks. As the case in the option-critic paper, they use different numbers for the navigation task and the pinball task. Their experiments showed that the performance is very sensitive to the number. Especially in real life applications, it is often very expensive to train multiple times in order to choose the parameters.
>
> >> Q5:  Why is the posterior (on $o$) conditioned only on the action history, and not state?
>
> A5: This is a design choice. We have a brief discussion on the open-loop controller and close-loop controller in the conclusion part. A close-loop controller would depend on state $s$. This could be a good direction for future study.
>
> >> Q6:  Additionally $o$ is being treated as a random variable through 3.2. So, what is $o$? Where is the randomness coming from?
>
> A6: The option in our formulation is a latent representation of a sequence of actions. We model the option following the definition of VAE, which treats the latent representation as a random variable. We should not have emphasized this too much, because the option does not have to be random in the TAIC framework. One can also use vanilla Auto-Encoders instead of VAE. It will also be an interesting study to analyze and compare the two.

---

> > ### Comment · AnonReviewer1 · 2019-11-15
> > **R1 Response to (1/3)**
> >
> > I thank the authors for their responses!
> >
> > Re A1: Understood, thanks.
> >
> > Re A2: It is still unclear to me that smoothness in particular is a desirable property. I would suggest rephrasing the statement 'smooth option space provides the RL algorithm with a better search space' to something like 'smooth option space allows interpolation between different options', or something to that affect. By my knowledge, we don't know whether a smooth or non-smooth option space is more useful for RL.
> >
> > Re A3: Understood -- this is a new framework for learning temporal abstractions. It's unclear why comparison can't be made to existing HRL methods, though; it is known that HRL can help speed up RL (see references from the review; Nachum et al. 2019, Levy et al. 2019, Sutton et al. 1999, and so on). However, contrasting the learning gains made by TIAC to something standard like the Option Critic would add clarity to the conditions under which TIAC is well suited to improve RL. As it stands, when would we expect to see TIAC improve over something like the Option Critic? One point is discussed (as in Q4/A4): the Option Critic requires tuning of the parameter specifying the number options. However, we could naturally not tune that parameter and still run the Option Critic (just pick a large number, for instance)---would we expect the Option Critic to outperform TIAC or not? Under what circumstances? Carrying out systematic experimentation to highlight when and why TIAC is effective would be helpful. Since the support offered for the method is empirical, I take it to be important to relate its performance to at least one other HRL (and specifically option) method. Perhaps the other reviewers can chime in here, though my reading is that R2 has a similar view.
> >
> > Re A5: Understood, thank you!

---

> > > ### Author Response · Authors · 2019-11-15
> > > **Author Response to Reviewer #1's Response (1/3)**
> > >
> > > Re: Re A2: Yes, your concern is right. The evidence of the correlation between smoothness and performance is not shown in this paper. We rephrased the statement in the updated version.
> > >
> > > Re: Re A3: Thanks for your comments. We think what you suggest is fair. Adding an HRL baseline will be our next step.

---

### Official Review · AnonReviewer3 · 2019-10-24
**Official Blind Review #3**

**Rating:** 3

**Review:**

Summary: This paper studies the hierarchical reinforcement learning (HRL) problem. It proposes a framework TAIC that learns temporal abstraction from past experience or expert demons without task-specific knowledge. The method is to formulate the problem by a temporal abstraction problem. That is, they assume that the action sequence is generated by a latent variable o. By regularizing the latent space by adding information-theoretic constraints, they are able to learn the representation. The paper later uses visualization to demonstrate the effectiveness of the learning.

I would think this paper is slightly below the borderline. It is an interesting method of encoding the option sequence by a continuous variable. Therefore, the action space becomes continuous rather than discrete. However, I found it not convincing why continuous option space is better than discrete ones. It appears to me that the experiment section does not provide a comparison with previous discrete option based methods as well.

Comments:
* 4th line of related work: Parr --> "Parr & Russel"
* Page 2, problem formulation: in beta(s,o), s is not defined. Maybe you can denote it as beta_o(.).
* It appears to be that the"option" is a sequence of actions? This can only happen in the deterministic environment. What will you do if applying pi does not give the same sequence of actions? For instance, from (s1,a1) -> (s2, a2), where s2 is generated from a random distribution, and a2 is based on s2.
* the paper is overlength

**Experience Assessment:**

I have published in this field for several years.

**Review Assessment: Checking Correctness Of Derivations And Theory:**

I assessed the sensibility of the derivations and theory.

**Review Assessment: Checking Correctness Of Experiments:**

I assessed the sensibility of the experiments.

**Review Assessment: Thoroughness In Paper Reading:**

I made a quick assessment of this paper.

---

> ### Author Response · Authors · 2019-11-11
> **Response to Reviewer #3**
>
> >> Q1: I found it not convincing why continuous option space is better than discrete ones. It appears to me that the experiment section does not provide a comparison with previous discrete option based methods as well.
>
> A1: Although the option we use in this paper is continuous, but our framework also supports discrete options.  We have not come to the conclusion of whether continuous or discrete options is more preferable. The main purpose of our method is trying to abstract high-level representations from the action sequences. In order to do so, we utilize the RVAE, which encodes sequential data into a continuous latent representation. It will be an interesting future direction to see how it works if we use discrete latent representation. For example, we can assume Bernoulli distribution instead of Gaussian distribution in the RVAE latent space, which will result in a discrete option. In this case, we can use DQN or other discrete RL solvers on top of it.
>
> One obvious tradeoff between continuous and discrete options is that the continuous option has more expressibility (theoretically, it can model an infinite number of action sequences uniquely); while training discrete options might be harder, but it can gigantically reduce the size of option space, which could benefit the RL training.
>
> >> Q2: It appears to be that the"option" is a sequence of actions? This can only happen in the deterministic environment. What will you do if applying pi does not give the same sequence of actions? For instance, from (s1,a1) -> (s2, a2), where s2 is generated from a random distribution, and a2 is based on s2.
>
> A2: The option in our formulation is a latent representation of a sequence of actions.
> Then the decoder D is responsible for decoding the option back into a sequence of action. The learning happens in two phases. First, the encoder E and decoder D are trained on sequences of actions with other networks (P and F) as regularization. Second, during the HRL training, the policy pi learns to output an option, which is decoded by the decoder D. The algorithm is not restricted to deterministic environments, because the policy pi learns to output accordingly with the state. In our experiments, the pi outputs a random variable, same as the setup in the PPO algorithm.
>
> >> Notations and typo.
> We have added the notation, and changed the typo.
>
> >> The paper is overlength
> The current manuscript is in the limit of ICLR, which is 10 pages. We are sorry for the extra effort required!

---

> > ### Comment · AnonReviewer3 · 2019-11-15
> > **An option is always a sequence of actions?**
> >
> > So an option is always a sequence of actions regardless of the states? How can you execute an option if in the middle some action is not available at a state?

---

> > > ### Author Response · Authors · 2019-11-15
> > > **Author Responseto Reviewer 3**
> > >
> > > Option in our definition is a latent representation of action sequence. There is an assumption that all the options are applicable to all states, as a common assumption also adopted by Option-critic, etc. We have defined multiple termination conditions, which allow detecting abnormal states and end the option earlier.

---

### Decision · Program_Chairs · 2019-12-19

**Decision:**

Reject

**Comment:**

This paper presents a novel hierarchical reinforcement learning framework, based on learning temporal abstractions from past experience or expert demonstrations using recurrent variational autoencoders and regularising the representations.

This is certainly an interesting line of work, but there were two primary areas of concern in the reviews: the clarity of details of the approach, and the lack of comparison to baselines. While the former issue was largely dealt with in the rebuttals, the latter remained an issue for all reviewers.

For this reason, I recommend rejection of the paper in its current form.